# Dynamic Fusion of Eye Movement Data and Verbal Narrations in Knowledge-Rich Domains

**Ervine Zheng**     **Qi Yu**[*]     **Rui Li**     **Pengcheng Shi**     **Anne Haake**
Rochester Institute of Technology
{mxz5733, qi.yu, rxlics, spcast, arhics}@rit.edu

## Abstract

We propose to jointly analyze experts' eye movements and verbal narrations to discover important and interpretable knowledge patterns to better understand their decision-making processes. The discovered patterns can further enhance data-driven statistical models by fusing experts' domain knowledge to support complex human-machine collaborative decision-making. Our key contribution is a novel dynamic Bayesian nonparametric model that assigns latent knowledge patterns into key phases involved in complex decision-making. Each phase is characterized by a unique distribution of word topics discovered from verbal narrations and their dynamic interactions with eye movement patterns, indicating experts' special perceptual behavior within a given decision-making stage. A new split-merge-switch sampler is developed to efficiently explore the posterior state space with an improved mixing rate. Case studies on diagnostic error prediction and disease morphology categorization help demonstrate the effectiveness of the proposed model and discovered knowledge patterns.

## 1   Introduction

Recent years have seen an increasing application of automatic computational systems in supporting humans in visual-based decision-making tasks. Machine learning models are applied to process large-scale data in the forms of images, videos, and texts for discovering statistical regularities and making predictions [1, 2]. However, human expertise is still essential in providing meaningful interpretations of the semantics for tasks in specialized domains, such as medicine, science, and security intelligence. Domain expertise, such as conceptual and perceptual skills, are usually developed through long-term training and practice. It allows human experts to perform better than fully automatic systems, which interpret images or videos solely based on low-level features [3, 4]. Therefore, it is beneficial to incorporate human behavioral data for visual-based tasks in knowledge-rich domains.

Modern technologies have made it possible to record human behavioral data [5, 6]. For instance, eye tracking measures the gaze and the motion of eyes to indicate how human perceptually processes images and audio recording digitally inscribes and re-creates human speeches as input for studying semantic conception. Analysis of eye gaze exposes cognitive processing at the level of visual perception, while verbal expression reflects semantic conception. These elements, both of which are significantly relevant to domain expertise, interact in visual-based decision-making process [7, 8].

In this paper, we propose to perform *dynamic multimodal knowledge data fusion* to synergize human domain expertise and statistical modeling, enabling them to tackle highly challenging visual-based tasks collectively. Inspired by psychological studies of important phases in humans' decision-making [9], we develop a phase-aware dynamic Bayesian nonparametric model that assigns latent knowledge patterns into key phases involved in complex decision-making. In particular, an expert's decision-making process is automatically partitioned into a sequence of latent decision phases, whose temporal

---

[*]Corresponding author

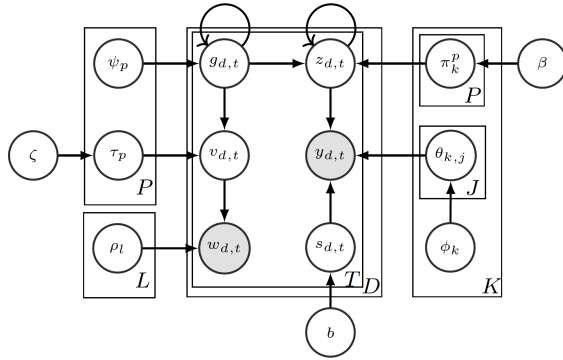

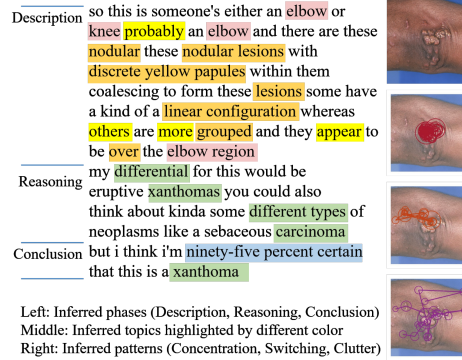

Figure 1: Graphical model of phase-aware knowledge fusion (hyper-parameters are omitted, and curved arrows denote first-order Markov transition; $L, J, K$ are potentially infinite; the notations for variables are summarized in the supplementary material)

Figure 2: An illustrative example of inferred latent knowledge patterns of physicians' verbal narrations and eye movements, and the latent phases explain experts' diagnostic decision making process.

dependency is captured by a Markov structure. We further model the cross-modal interactions of multimodal data by conditioning both perceptual behavior (as eye movement patterns in our case) and conceptual processing (as topics from verbal narrations) on the decision phases. As a result, the multimodal latent patterns are dynamically fused at the phase level by contributing different knowledge components to a specific decision stage.

To perform phase-aware fusion of eye movements, we integrate an *infinite hidden Markov model with a nested Dirichlet process mixture (iHMM-nDP)* to capture the spatiotemporal characteristics of eye movements. Since we aim to discover perceptual patterns common to a group of experts, commonly used models may lead to a large number of patterns with minor spatial/temporal variations. Hence, extensive post-processing is usually needed to group semantically similar patterns [10]. The proposed iHMM-nDP model addresses this issue by naturally forming a 3-level semantic hierarchy, including *state, component, and instantiation*, which capture main patterns, sub-patterns with minor spatial/temporal variations, and actual observations from individual experts. We further leverage the hierarchical Dirichlet process (HDP) model to perform phase-aware fusion of the verbal narrations. Phase-specific word topics are discovered that help explain the conceptual patterns conditioned on the same phase. As a result, the phase-aware fusion model reveals the relationship between eye movements and verbal narrations, creates knowledge-centered representations of data, and ultimately contributes to the understanding of experts' decision-making process. Figure 1 shows the overall graphical model. Finally, a new *Split-Merge-Switch (SMS) sampler* is developed to efficiently explore the posterior state space with an improved mixing rate.

Figure 2 illustrates how the proposed model explores experts' decision making process by visualizing patterns and topics learned from eye movements and verbal narrations, respectively. Each circle represents a location of visual fixation, and the radius is proportional to the duration. Three significant patterns are visualized in this example, including concentrating on primary abnormality, switching among several locations, and cluttering within a specific area [10]. The keywords from different latent topics are shown in different colors. Moreover, the proposed model automatically partitions the narration into three different decision phases. As can be seen, the narration starts from the description of low-level visual features of diseases, then goes through a reasoning process, and finally reaches a conclusion. The major contributions are summarized below:

- a phase-aware dynamic Bayesian nonparametric model to fuse experts' eye movements and verbal narrations in complex decision-making based on key decision phases.
- an iHMM-nDP model to extract perceptual patterns that summarize spatiotemporal regularities from eye movements through a three-level semantic hierarchy to capture the main patterns, the sub-patterns, and the observations of eye movements hierarchically; discovery of phase-specific topics that help explain the conceptual patterns as a result of fusing experts' verbal narrations.
- a fast mixing Split-Merge-Switch sampling algorithm to efficiently explore a potentially large latent state space due to nonparametric modeling and speed up hierarchical pattern discovery.

For evaluation, we present case studies on diagnostic error prediction and disease morphology categorization to demonstrate the effectiveness of the proposed model and discovered patterns.

## 2   Related Works

**Learning from Multimodal Data.** Multimodal machine learning aims to leverage data with multiple modalities for generating improved representation, relating data from one modality to another, identifying cross-modal relationships, transferring knowledge, or joining multiple modalities to perform predictions [11]. Our work aims to achieve most of the above goals.

Multi-kernel learning extends support vector machines (SVMs) to allow different kernels for different modalities. It can be used for data fusion in knowledge-rich domains such as disease prediction [12]. A drawback of multi-kernel learning is the high space complexity and slow convergence. Matrix decomposition can also be applied to data fusion, in which data is factorized as the product of matrices capturing shared features across modalities and matrices capturing the uniqueness of each modality by maximizing correlation or minimizing squared errors and divergences [13, 14]. Bayesian graphical models provide another way of data fusion [15, 16]. Recently, deep learning models have been used for fusing temporal multimodal information [17, 18, 19, 20, 21]. They usually show good performance by learning complex decision boundaries [11]. However, for problems in knowledge-rich domains, interpretability of latent patterns is usually essential while the data for model training may be inadequate, which limits the applicability of most deep learning models.

**Samplers for Bayesian Nonparametric Models.** The HMM and linear dynamical systems are typical models for analyzing sequential data, where hidden states relate to each other through a Markov process [22]. Those dynamic models can be extended to Bayesian non-parametric counterparts using HDP or hierarchical Beta processes (HBP) [23, 24, 25]. The posterior inference for HDP based HMM can be performed through Markov Chain Monte Carlo (MCMC) samplers, such as Gibbs sampling, blocked sampling, and Beam sampling [26, 27], or through variational inference, which makes a truncation of potentially infinite states [28]. MCMC-based samplers provide an asymptotically exact inference but usually suffer slow mixing, because the incremental updates of state assignment conditioned on the previous observations and model hyper-parameters may be trapped in local optima. To address this issue, the split-merge algorithm is proposed to change the state assignments over a group of observations in a single move, which allows efficient exploration of a state space [29, 30, 31, 32]. We make extensions by proposing a *split-merge-switch sampler* to perform three-level hierarchical clustering of experts' eye movement patterns in a non-parametric fashion.

## 3   Multimodal Knowledge Data Description

Two data elicitation experiments were conducted chronologically in prior works [33, 34] by using a repository of dermatological images as visual stimuli. We chose dermatology, as it is a visually based medical specialty that requires specific and comprehensive expertise. The 48 images used in the first experiment (Experiment I) represented a wide range of dermatology diagnoses, while the 30 images in the second experiment (Experiment II) focused on a few categories of diagnoses, each with more image instances. There were 16 participating physicians in the first experiment, and 29 in the second. They volunteered to participate with monetary compensation. The headwear Senso-Motoric eye-tracking devices with 50 Hz sampling rate automatically record the fixations and saccades. Experiments were conducted in an eye-tracking laboratory. Dermatological images were presented on-screen at a resolution of 1680x1050 pixels. Participants viewed the images binocularly at a distance of 60 cm. They were instructed to describe each image and their thought processes towards diagnosis. Their diagnostic decisions were evaluated by a group of senior experts. IRB approval has been received before the data collection experiments were conducted.

**Eye movement data.** As a channel for visual content perception, physicians' eye movements were recorded by eye trackers. Two important events commonly studied in eye movement research are *saccades* and *fixations*. Fixations, when the gaze is maintained on a location, are described by location, duration, and in some cases pupil dilation. The high-speed eye movements between two fixations are saccades and are characterized by amplitude, the length in degrees of visual angle, and the velocity in degree per second. Since eye movement sequences are spatiotemporal, they can be best represented as time series. **Verbal narration data.** All verbal narrations were recorded and transcribed as sequences of word tokens and time-stamps using the speech analysis tool Praat [35].

# 4 Dynamic Multimodal Knowledge Fusion

In this section, we present the phase-aware dynamic Bayesian nonparametric model that analyzes experts' eye movements and verbal narrations jointly and hierarchically.

## 4.1 Phase-Aware Multimodal Knowledge Data Fusion

We assume the decision-making process is comprised of *three major phases*: description, reasoning, and conclusion. Extension to more phases is straightforward. A first-order Markov structure is used to capture the transition of phases. Let $\psi$ denote the transition matrix and a Dirichlet prior is placed to each row $\psi_p = Dir(\omega_0)$. For sequence $d$, its latent phase $g_{d,t}$ at time $t$ is drawn from

$$g_{d,t} \sim \text{Multi}(\psi_{g_{d,t-1}}) \tag{1}$$

where the phase assignment to each observation is inferred from the data.

A sequence of latent phases governs a physician's eye movements and verbal narration. Eye movement is primarily a visual information gathering process, which facilitates physicians in decision-making. Physicians' eye movement patterns reveal different characteristics at different phases, from looking around to collect general information to fixate at disease areas to extract detailed information. Verbal narration is essentially the decision-making process "spoken-aloud" by experts. Topics from verbal narrations vary at different phases and capture important keywords of the corresponding phases. Integrating eye movements with verbal narrations will help improve the understanding of a complex decision-making process because the underlying *perceptual* (i.e., eye movement) and *conceptual* (i.e., topics) patterns are expected to capture distinct but complementary domain expertise. It motivates us to explicitly model the conditional dependency of both topics and eye movement patterns on the decision phases, and use the phases as a basis to fuse the two knowledge modalities, leading to the discovery of *phase-specific* topic distribution and the transition probability of eye movement patterns.

In summary, the phase-aware fusion offers two unique benefits: (1) The decision phases provide further evidence (though its density function) in addition to the eye movements and verbal narrations to strengthen the significant patterns and weaken the noisy ones. (2) Interesting behavior becomes interpretable through both the hierarchical and parallel interactions among decision phases, eye movement patterns, and word topics. Figure 1 shows the graphical model of the overall dynamic data fusion process. For sequence $d$ and time step $t$, the latent phase $g_{d,t}$ has a Markov transition structure, as denoted by the curved arrow. Both topic assignment $v_{d,t}$ and eye movement pattern assignment $z_{d,t}$ are conditioned on $g_{d,t}$. All the notations are summarized in Table 3 of Appendix A.

## 4.2 Fusion of Eye Movements

Modeling eye movements is challenging because the characteristics of eye movements may vary a lot for different physicians. For example, physicians may unintentionally move eyes and heads in experiments. Furthermore, head-wear eye-trackers may have instrumental and calibration errors at different trails. To address the challenges, a model needs to be able to discover semantically coherent patterns while accommodating the variety and being robust to noises.

We propose an iHMM-nDP model to extract perceptual patterns that summarize spatiotemporal regularities from eye movements through *a three-level semantic hierarchy* to capture the main patterns, sub-patterns, and observations of eye movements hierarchically. In particular, we use the latent *states* in the iMM to model that main patterns (e.g., concentrating on a small area, or switching between two areas). Each state is comprised of a mixture of components, each of which captures a fine-grained sub-pattern (e.g., multiple concentration patterns characterized by different fixation duration and area). By modeling the states (main patterns) and mixture components (sub-patterns), we essentially perform *dynamic hierarchical clustering* of eye movements in a *non-parametric* fashion.

Let $\boldsymbol{y}_{d,t} \in \mathbb{R}^D$ denote the vector representation of eye movements in sequence $d$ at time $t$, $z_{d,t}$ be the corresponding state assignment, and $s_{d,t}$ be the mixture component assignment.

$$(\boldsymbol{y}_{d,t}|_{z_{d,t}=k,s_{d,t}=j}) \sim \mathcal{N}(A_{k,j}, \Sigma_{k,j}) \tag{2}$$

where $\boldsymbol{y}_{d,t}$ is assumed a normal variable, and $(A_{k,j}, \Sigma_{k,j})$ is the corresponding parameter of the sub-pattern indexed by $(k, j)$, $S_0$ is the scale matrix and $d_0$ is the degree of freedom. We use a thermal diffusion process on eye gaze points in each period to generate 2-dimensional attention maps. Those maps encode visually attended areas by physicians, where the locations that are fixated by

physicians are assigned high values, and the locations far away are assigned low values. We then apply shifting and rotation so that the gravity center aligns with the map's center, and the direction with the largest variance is horizontal. Then the maps are shrunk, compressed using 2D2D-PCA [36] and flattened to generate the representation of eye movements.

Each sub-pattern associates with a unique set of coefficients $\{A_{k,j}, \Sigma_{k,j}\}$. Since the number of states is unknown, the iHMM model infers it by placing an HDP prior onto the state transitions [24], where each DP governs the transition probability for each group of states that the current state can transit to. All groups share a base distribution, which is another DP so that the same set of states can be reachable from any current state. Let $H$ denote the global base measure parameter $\phi_k$ is drawn from, and $G_0$ denote the first level DP, representing a space of potential states:

$$G_0 = \sum\nolimits_{k=1}^{\infty} \beta_k \delta_{\phi_k}, \quad \boldsymbol{\beta} \sim \text{GEM}(\gamma), \quad \phi_k \sim H \tag{3}$$

where $\boldsymbol{\beta} = (\beta_1, ...\beta_k, ...)'$ follows GEM distribution [37]. The actual form of $\phi_k$ is defined in below.

To achieve phase-aware fusion of eye movements, *one key extension* from the classical HDP-HMM model is to make the second level DP *phase dependent*:

$$G_k^p = \sum\nolimits_{k'=1}^{\infty} \pi_{kk'}^p \delta_{\phi_{k'}} \quad z_{d,t}|_{z_{d,t-1}=k} \sim \pi_k^p \tag{4}$$

where $\pi^p$ is the phase-specific transition matrix, and its $k$-th row, $\pi_k^p \sim DP(\alpha, \boldsymbol{\beta})$ with $\alpha$ as a concentration parameter. $\pi_{kk'}^p$ denotes the transition probability from state $k$ to $k'$ at phase $p$.

Another *key extension* is that we couple each hidden state with nested Dirichlet process (nDP) to handle its emission process. Different from a conventional nDP, where all mixture components share a global base measure, we aim to cluster mixture components with small intra-cluster variety. We propose a *modified nDP*, where $H_k$, the base measure of components in state $k$, is state-specific:

$$G_k^* = \sum\nolimits_{j=1}^{\infty} b_{k,j} \delta_{\theta_{k,j}}, \quad \boldsymbol{b} \sim \text{GEM}(\gamma^*), \quad \theta_{k,j} \sim H_k(\phi_k) \tag{5}$$

where $\theta_{k,j} = (A_{k,j}, \Sigma_{k,j})$ is defined below. We place a normal prior for $\phi_k \sim \mathcal{N}(A_0, U_0)$, where hyper-parameter $A_0$ is the mean and $U_0$ is the covariance. For state $k$ component $j$, a normal Inverse Wishart prior $\text{NIW}(A, \Sigma|S_0, d_0, \kappa, \phi_k)$ is placed on $\theta_{k,j} = (A_{k,j}, \Sigma_{k,j})$:

$$A_{k,j} \sim \mathcal{N}(\phi_k, \Sigma_{k,j}/\kappa) \quad \Sigma_{k,j} \sim \text{IW}(S_0, d_0) \tag{6}$$

where $\kappa$ is the scaling parameter.

## 4.3 Fusion of Verbal Narrations

To perform phase-aware fusion of verbal narrations, the model enforces a *phase-specific* topic distribution. In particular, the corpus-level topics $M_0$ are generated as follows:

$$M_0 = \sum\nolimits_{l=1}^{\infty} \zeta_l \delta_{\rho_l}, \ \rho_l \sim \text{Dir}(\omega), \ \boldsymbol{\zeta} \sim \text{GEM}(\xi) \tag{7}$$

Each phase has a unique topic distribution. For phase $p$, a document-level $M_p$ is generated from $M_0$:

$$M_p = \sum\nolimits_{l=1}^{\infty} \tau_{p,l} \delta_{\rho_l}, \quad \tau_p \sim \text{DP}(a, \boldsymbol{\zeta}) \tag{8}$$

For time step $t$, topic assignment $v_{d,t}$ and word $w_{d,p,t}$ are:

$$v_{d,t} \sim \text{Multi}(\tau_{g_{d,t}}), \quad w_{d,t} \sim \text{Multi}(\rho_{v_{d,t}}) \tag{9}$$

By grouping words from each phase, the model encourages the inferred topics to capture the keywords associated with different decision phases (e.g., differential, final, diagnosis).

## 4.4 Split-Merge-Switch (SMS) Sampling

The nonparametric nature of the model coupled with its multi-level hierarchical structure that fuses complex multimodal data makes the posterior inference extremely complex and time-confusing. To this end, we develop a split-merge-switch sampler to achieve fast inference.

Traditional MCMC samplers for Bayesian nonparametric mixture models, such as Gibbs sampler, may be trapped in an inappropriate clustering of data and result in slow mixing. The primary reason

is that Gibbs samplers perform a single-site update for latent pattern assignment. Beam sampling [27] and block sampling [38] mitigate the problem by using a forward-backward procedure to update the pattern assignments for a whole sequence. However, the problem of slow mixing may still emerge when there are a large number of sequences. The proposed SMS sampler provides a solution by changing the pattern assignments over a group of observations in a single move. We consider three types of proposals, namely *split, merge*, and *switch*. The three proposals are *mutually exclusive*. Each proposal is evaluated using a Metropolis-Hasting acceptance ratio. If accepted, the proposal is implemented; if not, we ignore the proposal and proceed to sample other variables.

The split proposal suggests splitting a mixture component into two within the same state. Let $S$ denote the indices of events assigned to state $k$ and component $j$. First, a pair of indices $[(d_i, t_i), (d_o, t_o)]$ is randomly selected from $S$ and serve as anchors. Then we remove them from $S$ and form singleton sets $S_i = \{(d_i, t_i)\}$ and $S_o = \{(d_o, t_o)\}$. For each $(d, t) \in S$, we sequentially add it to $S_i$ with

$$p((d,t) \in S_i | S_i, S_o, y_{d,t}) = \frac{|S_i| \int f(y_{d,t}|\theta) dH_{S_i}(\theta)}{|S_i| \int f(y_{d,t}|\theta) dH_{S_i}(\theta) + |S_o| \int f(y_{d,t}|\theta) dH_{S_o}(\theta)} \tag{10}$$

where $H_{S_i}(\theta)$ is $S_i$'s posterior distribution of $\theta$. Otherwise, we add it to $S_o$.

**Lemma 1.** *For a split proposal $Split(S) = (S_i, S_o)$ where $S = \{(d,t)|z_{d,t} = k, s_{d,t}^{old} = j\}$, $S_i = \{(d,t)|z_{d,t} = k, s_{d,t}^{new} = j_1\}$, and $S_o = \{(d,t)|z_{d,t} = k, s_{d,t}^{new} = j_2\}$, the following acceptance ratio satisfies the detailed balance*

$$a(\boldsymbol{\eta}^{old}, \boldsymbol{\eta}^{new}) = \min[1, \frac{p(\boldsymbol{\eta}^{new})p(\boldsymbol{y}|\boldsymbol{\eta}^{new})p(\boldsymbol{\eta}^{old}|\boldsymbol{\eta}^{new})}{p(\boldsymbol{\eta}^{old})p(\boldsymbol{y}|\boldsymbol{\eta}^{old})p(\boldsymbol{\eta}^{new}|\boldsymbol{\eta}^{old})}] \tag{11}$$

*with*

$$\frac{p(\boldsymbol{\eta}^{old}|\boldsymbol{\eta}^{new})}{p(\boldsymbol{\eta}^{new}|\boldsymbol{\eta}^{old})} = 1/\prod_{(d,t)} p((d,t)|S_i, S_o, y_{d,t}), \quad \frac{p(\boldsymbol{\eta}^{new})}{p(\boldsymbol{\eta}^{old})} = \gamma^* \frac{(|S_i| - 1)!(|S_o| - 1)!}{(|S| - 1)!} \tag{12}$$

*where $\boldsymbol{\eta}^{old}$ is the old assignments, $\boldsymbol{\eta}^{new}$ is the proposed new assignments, and $\gamma^*$ is defined in* (5).

The merge proposal suggests merging two mixture components from the same state into a single one. It is essentially the reverse of a split proposal, and the acceptance ratio is calculated in a reversed way.

The switch proposal suggests moving some mixture components from one state and adding them to another state as additional mixture components, while keeping the grouping of elements within each mixture unchanged. We further consider two cases, namely switch1, where some mixture components from a pattern $k_1$ are moved to a new pattern $k_2$, and switch2, where all mixture components from a pattern $k_3$ are moved to an existing pattern $k_4$.

**Lemma 2.** *For a switch proposal $Switch1(S_1) = (S_2, S_3)$ where $S_1 = \{(d,t)|z_{d,t}^{old} = k_1, s_{d,t}^{old} = 1 : J\}$, $S_2 = \{(d,t)|z_{d,t}^{new} = k_2, s_{d,t}^{new} = 1 : j_2\}$ and $S_3 = \{(d,t)|z_{d,t}^{new} = k_1, s_{d,t}^{new} = 1 : J - j_2\}$, the following acceptance ratio satisfies the detailed balance in Eq 11 with*

$$\frac{p(\boldsymbol{\eta}^{old}|\boldsymbol{\eta}^{new})}{p(\boldsymbol{\eta}^{new}|\boldsymbol{\eta}^{old})} = 1/\prod_{(d,t)} p((d,t)|S_2, S_3, y_{d,t}), \quad \frac{p(\boldsymbol{\eta}^{new})}{p(\boldsymbol{\eta}^{old})} = \frac{p(\boldsymbol{s}^{new}|\boldsymbol{z}^{new})}{p(\boldsymbol{s}^{old}|\boldsymbol{z}^{old})} \frac{p(\boldsymbol{z}^{new})}{p(\boldsymbol{z}^{old})} \tag{13}$$

*and*

$$\frac{p(\boldsymbol{s}^{new}|\boldsymbol{z}^{new})}{p(\boldsymbol{s}^{old}|\boldsymbol{z}^{old})} = \prod_{i=1}^{|S_2|}(\gamma^* + i - 1)\prod_{i=1}^{|S_3|}(\gamma^* + i - 1)/\prod_{i=1}^{|S_1|}(\gamma^* + i - 1)$$

$$p(\boldsymbol{z}^{old}) = \gamma^K \beta_{u_1}\beta_{u_2}...\beta_{u_k}\prod_{k=1}^{K}\prod_{k'=1}^{K}\prod_{i=1}^{n_{kk'}}(\gamma\beta_{k'} + i - 1)/\prod_{k=1}^{K}\prod_{i=1}^{n_k}(\gamma + i - 1) \tag{14}$$

*where $\beta_{u_k} = 1 - \sum_{i}^{k-1}\beta_i$ And $p(\boldsymbol{z}^{new})$ can be calculated similarly.*

The switch2 proposal is essentially the reverse of a switch1 proposal, and the acceptance ratio is calculated in the reversed way.

The calculation of $\frac{p(\boldsymbol{\eta}^{old}|\boldsymbol{\eta}^{new})}{p(\boldsymbol{\eta}^{new}|\boldsymbol{\eta}^{old})}$ is performed using sequential allocation similar to Eq 10, but we allocate one mixture at a time, instead of one data point individually. Since the number of mixture is far less than the number of data points, the allocation can be performed quickly. Notice that the switch also incurs the change of corresponding $\boldsymbol{\theta}$'s conditional dependency on $\phi$, because the component is assigned to a new state.

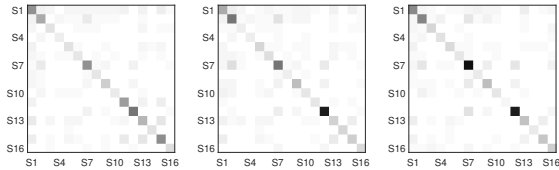
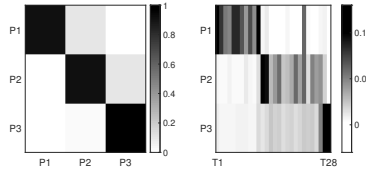

Figure 3: Eye movement state transition counts at Description (Left), Reasoning (Middle), and Conclusion Phases (Right)

Figure 4: Visualization of phase transition matrix (Left) and topic distribution of each phase (Right) in Experiment I

The *entire posterior inference* starts by sampling the latent decision phase (detailed in Appendix B.3). It then simultaneously samples latent topics of verbal narrations (Appendix B.2) and latent state of eye movements and its mixture component (Appendix B.1). Finally, the states and their mixture components are further updated using the SMS sampler (Appendix B.5). The whole process is also summarized in Algorithm 1 in the Appendix.

## 5 Results and Discussion

In this section, we first present the discovered patterns along with their cross-modal interactions by applying the proposed dynamic data fusion model to the data collected from the two experiments, as described in Section 3. We then present two case studies on diagnostic error prediction and disease morphology categorization, respectively, to further demonstrate the proposed data fusion model's effectiveness and the discovered knowledge patterns.

### 5.1 Discovery of Perceptual Patterns

We consider eye movement fixation events as observation units. All observation units have three data fields, including the x-y coordinates of fixation and its duration. We interpret the discovered eye movement patterns as 1) *Concentration* pattern characterized by fixations with long duration and saccade with a small amplitude. It usually associates with primary abnormalities. 2) *Switching* pattern characterized by fixations with short duration and saccade with large amplitude. It usually associates with two abnormalities; 3) *Clutter* pattern characterized by fixations with short duration and saccade with a large amplitude. It usually associates with multiple abnormalities.

Illustrative examples are provided in Table 1. Our algorithm discovered sixteen significant states of eye movement patterns from Experiment I and II. Three patterns are visualized. The line segments show the change of coordinates in saccade, and the circles show the fixation locations. We interpret the first row as Concentration, the middle row as Switching, and the bottom row as Clutter.

Figure 3 visualizes the transition matrix of eye movement states of Experiment I using grey-scale mapping. The diagonal line corresponds to the self-transitions of states. We observe higher self-transition of S7 (concentration) at the conclusion phase, which indicates that physicians' eye movements are more stable upon conclusion. There are more non-concentration transitions at the description, such as the self transition of S13 (clutter) and the transition from S13 to S15 (clutter), indicating that experts change eye movement states more frequently in order to gather general information from the entire image. Additional examples and interpretations are provided in Appendix C.

### 5.2 Discovery of Conceptual Patterns

In Table 1, we show the partition of verbal narrations based on the inferred phase assignments, and highlight the informative words from inferred topics. We also study the transition pattern of phases and the relationship between phases and topics, aiming to gain more in-depth insight into experts' problem solving and decision-making processes. In particular, we visualize the occurrence of phase transition as well as the topic distribution of each phase using grey-scale mapping as shown in Figure 4. Some interesting and intuitive observations are provided as follows: First, each decision phase has a strong self transition, and there are moderate occurrences of phase transition from description (P1) to reasoning (P2), and from reasoning to conclusion (P3). Second, each phase is associated with a unique set of topics (e.g., the conclusion phase is closely related to the last three topics (T26-T28) in experiment I. More details about the inferred topics and their top words are summarized in Appendix C.

Table 1: An illustrative example of inferred phases, topics and eye movement patterns from different cases: informative words with high weights in the corresponding topics are highlighted in colors; each sub-figure corresponds to a fine-grained eye movement pattern. (Due to space limit, we selectively visualize the topics and the patterns)

| | Participant 1 Image 1 | Participant 1 Image 2 | Participant 2 Image 1 | Participant 2 Image 2 |
|---|---|---|---|---|
| *Narration* | ——Description—— so some erythematous scaly almost annular plaques inner thigh like of a female uh diagnoses on the or the ——Reasoning—— differential tinea corporis myconsis fungoides um erythema annulare centrifugum or eac probably ——Conclusion—— favor mycosis fungoides but that's only like thirty percent certainty | ——Description—— the face you have some erythematous patches extending on the nose and malar region of the cheek sparing the crura like there's a little erythema as well at the entrance to the left nare ——Conclusion—— number one thought would be seborrheic dermatitis with seventy-five percent certainty | ——Description—— here erythematous patches with central clearing and scale on the lower extremity ——Reasoning—— the differential includes uh psoriasis nummular eczema tinea eac mycosis fungoides and the ——Conclusion—— diagnosis is eczema with fifty percent certainty | ——Description—— there's erythema and waxy scale on the nose nostril and surrounding the uh nasolabial fold ——Reasoning—— the differential includes uh seborrheic dermatitis atopic dermatitis uh lupus rosacea and ——Conclusion—— my diagnosis is seborreheic dermatitis with eighty-five percent certainty |
| *Eye Movement* | | | | |
| Concentration |  |  |  |  |
| Switching |  |  |  |  |
| Clutter |  |  |  |  |

## 5.3 Prediction of Diagnosis Correctness

As can be seen from the above analysis of our modeling results, the discovered knowledge patterns show strong links to humans' decision-making process. Therefore, these patterns may serve as a useful vehicle to detect potential diagnosis errors. In this set of experiments, the diagnostic decisions made by participating physicians were evaluated by a group of senior experts, and the correctness of diagnosis are labeled as correct, incorrect and partially correct. We use the patterns discovered by the proposed model to train an L1-regularized logistic regression for predicting diagnostic correctness.

The following baselines are implemented for comparison: 1) *Modeling eye movements only*: Mixture auto-regressive model (MAR) [38]; 2) *Modeling verbal narrations only*: LDA [39] and hidden Markov topic model (HMTM) [40]. 3) *Multimodal fusion*: LDA-based Multimodal Categorization (LDAMC) [16]. 4) *Ensemble method*: Proposed+LDA. In most cases, the proposed model or the ensemble method achieves best performance, indicating that physicians usually reveal informative clues in their behavior before making a correct diagnosis.

## 5.4 Prediction of Disease Morphology

The distribution and arrangements of lesions may guide diagnostic decisions because many skin abnormalities have a specific configuration, which is an important cue of correct diagnosis. Such configuration naturally corresponds to physicians' eye movements. Therefore, we try to use inferred eye movement patterns and verbal narration topics for discovering those configurations. The meaning of the visual features and their functional relations are unveiled by experts' domain knowledge, leading to disease morphology categorization at the semantic level and finally assisting diagnostic decision making. To demonstrate the usefulness of the discovered knowledge patterns, we use them as inputs to a regularized logistic regression to predict the disease morphology as one of the following types: *Solitary (Sol)*: a solitary lesion as primary abnormality; *Symmetry (Sym)*: symmetrically distributed lesions; *Multiple Morphologies (MM)*: lesions of different morphologies with one lesion as primary abnormalities and others as secondary ones; *High-Density Lesions (HDL)*: scattered or

Table 2: Prediction of Diagnosis Correctness (Left) and Disease Morphology (Right) (Accuracy %)

| | Experiment I | Experiment II | | Experiment I | Experiment II |
|---|---|---|---|---|---|
| MAR | 63.4±3.9 | 55.4±4.0 | MAR | 74.5±3.5 | 67.9±3.6 |
| LDA | 65.5±3.1 | 63.0±3.5 | LDA | 72.1±3.4 | 65.4±3.4 |
| HMTM | 64.7±3.7 | 63.3±3.8 | HMTM | 61.5±3.8 | 58.0±3.9 |
| LDAMC | 67.2±3.8 | 62.0±3.5 | LDAMC | 79.9±3.7 | 65.7±3.8 |
| Proposed | 68.8±3.5 | 64.1±3.7 | Proposed | 87.5±2.9 | 78.7±3.1 |
| Proposed+LDA | 69.1±3.2 | 64.5±3.6 | Proposed+LDA | 87.3±2.7 | 78.8±2.8 |

clustered lesions. Table 2 shows the comparison results, where the proposed model or the ensemble method achieves the best performance.

## 5.5 Experiment on Synthetic Data for the Split-Merge-Switch Sampler

For illustration purposes, we study the SMS sampler using synthetic data where the structure of the latent states is known in advance, which serves as ground-truth for evaluation. The synthetic data set is generated through hierarchical Gaussian mixtures with sequential dependency. In particular, the latent state structure consists of 4 states (i.e., main patterns), each of which has 4 sub-patterns. The main patterns are centered at [4.5,4.5], [-4.5,4.5], [-4.5,-4.5], and [4.5,-4.5] respectively. The centers of the sub-patterns slightly deviate from their corresponding main pattern's center. We initialize the number of main patterns $K = 2$ and the number of sub-patterns $J = 2$, and train an iHMM-nDP model with or without the SMS sampler. Additional details are provided in the Appendix D.

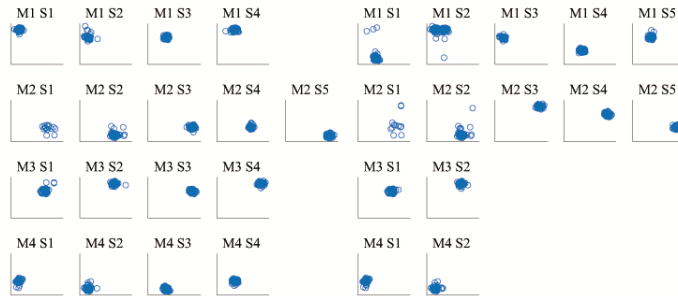

Figure 5: Visualization of inferred sub-patterns with and without SMS sampler (Left and Right)

The inferred sub-patterns are plotted in Figure 5. Each row in a sub-figure corresponds to a main pattern, and each column corresponds to a sub-pattern. The model with the SMS sampler correctly discovered four main patterns and almost all the sub-patterns, as shown in the left sub-figure. The model without the SMS sampler discovered four main patterns, but many sub-patterns are assigned incorrectly, as shown in the right sub-figure. For example, the main pattern 1's sub-pattern 4 (M1S4) should be assigned to main pattern 4. The results indicate that the SMS sampler contributes to the fast mixing rate and better hierarchical clustering results.

## 6 Conclusions

In this paper, we present a phase-aware dynamic Bayesian non-parametric model that jointly analyzes experts' eye movements and verbal narrations involved complex decision-making. By leveraging the conditional dependency of both perceptual and conceptual patterns on the key decision phases, multimodal knowledge data is naturally fused at the phase level. A novel iHMM-nDP model performs dynamic hierarchical clustering of noisy and highly variant eye movement events in a non-parametric fashion to discover an optimal number of main perceptual patterns along with their supporting sub-patterns. The phase-specific topics discovered as a result of fusing verbal narrations help explain the main perceptual patterns to ensure model interpretability. A fast mixing SMS sampler is developed to achieve efficient posterior inference. The usefulness of the discovered knowledge patterns is further demonstrated through real-world case studies. In this work, we study knowledge data from experts who are trained professionals. They analyze the images in a systematic process, and their verbal descriptions usually follow certain schemes. A future direction is to make the model more robust to cases where careless practitioners do not follow such schemes.

## Acknowledgement

This research was partially supported by NSF IIS award IIS-1814450 and ONR award N00014-18-1-2875. Any opinions, findings, and conclusions or recommendations expressed in this paper are those of the authors and do not necessarily reflect the official views of any funding agency. We would like to thank the participating physicians, the reviewers, and Logical Images, Inc. for images.

## Broader Impact

The need to explore elements involved in human knowledge-based cognitive processing and fuse them with machine intelligence, empowered through computational processing of large-scale complex data, has been recognized by a wide spectrum of specialized domains, such as medicine, science, social psychology, security intelligence, and more. This work will provide both theoretical underpinning and empirical evaluation of infusing human expertise into the design of computing systems, enabling them to collectively tackle highly challenging tasks in specialized domains that neither could individually perform to satisfaction. The research can be broadly applicable to diverse knowledge-rich domains, where the synergy of human and machine intelligence is essential to tackle highly complex computational tasks.

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
