[Supplementary Material]

# Appendix

**Appendix outline.** The appendix includes four main sections. In **Appendix** A, we summarize all the important notations of variables used in the main paper. In **Appendix** B, we provide details of the posterior inference of parameters and the derivation of the acceptance rate for the SMS sampler. The entire inference process is summarized in **Algorithm** 1. In **Appendix** C, we provide additional details of experiments, including the experiment settings, interpretation of inferred phases and cross-modal interactions, inferred topics and patterns. In **Appendix** D, we present ablation studies on the SMS sampler with qualitative and quantitative analysis.

## A   Summary of Notations

Table 3 summarizes the notations of major variables.

Table 3: Summary of Notations

| Notation | Explanation of notation |
|---|---|
| **Variables about Phases** | |
| $p$ | index of phases |
| $d$ | index of observation sequences |
| $t$ | index of time steps |
| $\psi$ | transition matrix of phases |
| $\omega_0$ | parameter of Dirichlet prior for phase transition |
| $g_{d,t}$ | phase assignment for narration $d$ step $t$ |
| **Variables about Verbal Narrations** | |
| $l$ | index of topics |
| $\zeta$ | 1st-level stick breaking partition for topics |
| $\tau_p$ | 2nd-level stick breaking partition for topics |
| $\xi$ | 1st-level DP concentration parameter for topics |
| $a$ | 2nd-level DP concentration parameter for topics |
| $\omega$ | parameter of Dirichlet prior for word distribution in topics |
| $\rho_l$ | parameter of word distribution for topic $l$ |
| $v_{d,t}$ | topic assignment for narration $d$ step $t$ |
| $w_{d,t}$ | word observation for narration $d$ step $t$ |
| **Variables about Eye Movements** | |
| $k$ | index of eye movement states |
| $j$ | index of eye movement mixture components |
| $\beta$ | 1st-level stick partition for eye movement states |
| $\gamma$ | 1st-level DP concentration parameter for states |
| $\alpha$ | 2nd-level DP concentration parameter for states |
| $b_k$ | stick partition of mixture components for state $k$ |
| $\gamma^*$ | DP concentration parameter for mixture components |
| $y_{d,t}$ | eye movement observation for sequence $d$ step $t$ |
| $z_{d,t}$ | latent state for sequence $d$, step $t$ |
| $\pi^p$ | transition matrix for phase $p$ |
| $A_{k,j}$ | coefficient for state $k$ component $j$ |
| $\Sigma_{k,j}$ | covariance matrix for state $k$ mixture component $j$ |
| $\theta_{k,j}$ | parameter for state $k$ component $j$ |
| $\phi_k$ | parameter of state $k$ |
| $H_k$ | state-specific measure for $\theta_{k,j}$ for any component $j$ |
| $H$ | global base measure for $\phi_k$ for any $k$ |
| $S_0$ | prior of scale matrix of IW distribution |
| $d_0$ | prior degree of freedom of IW distribution |
| $A_0$ | prior mean of $\phi$ |
| $U_0$ | prior covariance of $\phi$ |
| $D$ | dimensionality of $y_{d,t}$ |
| $n_{kk'}$ | counts of transition from pattern $k$ to pattern $k'$ |
| $u_{d,t}$ | auxiliary variable for slice sampling |

# B Details of Posterior Inference

In this section, we provide the details of the posterior inference for latent variables, including phases, topics, and eye movement patterns, and present the split-merge-switch sampler.

## B.1 Sampling Latent Variables of Eye Movements

### B.1.1 Sampling $z_{d,t}$ and $s_{d,t}$

Given the phase assignment $g_{d,t}$, we sample the eye movement state assignment $z_{d,t}$ and mixture component assignment $s_{d,t}$. Use the chain rule, $(z_{d,t}, s_{d,t})$ can be sampled from

$$p(z_{d,t}, s_{d,t}|\boldsymbol{y}_d, \boldsymbol{z}^{-dt}, \boldsymbol{s}^{-dt}, \boldsymbol{\pi}, \boldsymbol{\phi}, \boldsymbol{\theta}, g_{d,t}) \propto \quad p(s_{d,t}|z_{d,t}, \boldsymbol{y}_d, \boldsymbol{s}^{-dt}, \boldsymbol{\theta})p(z_{d,t}|\boldsymbol{y}_d, \boldsymbol{z}^{-dt}, \boldsymbol{s}, \boldsymbol{\pi}, \boldsymbol{\phi}, \boldsymbol{\theta}, g_{d,t}) \tag{15}$$

where $\boldsymbol{z}^{-dt}$ denotes all state assignments except for $z_{d,t}$, and $\boldsymbol{s}^{-dt}$ denotes all mixture component assignments except for $s_{d,t}$. The first term of Eq (15) can be estimated using

$$p(s_{d,t} = j|z_{d,t} = k, \boldsymbol{y}_d, \boldsymbol{s}^{-dt}, \boldsymbol{\theta}) \propto \begin{cases} \dfrac{n_{k,j}^{-t}}{\gamma^* + n_{k.}^{-t}}p(y_{d,t}|\theta_{k,j}) & j \leq J_k \\ \dfrac{\gamma^*}{\gamma^* + n_{k.}^{-t}}p(y_{d,t}|\phi_k, S_0, d_0) & j = J_k + 1 \end{cases} \tag{16}$$

where $J_k$ is the number of instantiated components for state $k$, and $n_{k,j}^{-t}$ is the counts of observations assigned to state $k$ component $j$ not including the observation at time step $t$.

To estimate the second term, we follow the Beam sampling algorithm proposed by [27]. In particular, an uniformly distributed auxiliary variable $u_{d,t}$ is introduced with probabilistic density

$$p(u_{d,t}|z_{d,t-1}, z_{d,t}, g_{d,t} = p, \pi^p) = \quad \frac{1}{\pi_{z_{d,t-1}, z_{d,t}}^p}\delta(0 < u_{d,t} < \pi_{z_{d,t-1}, z_{d,t}}^p) \tag{17}$$

where $\delta(.)$ is an indicator variable.

The Beam sampling is an extension to the slice sampling, and the auxiliary variable $u_{d,t}$ serves as a slice. It partitions $\pi_{z_{d,t-1}, z_{d,t}}^p$, so that a finite number of states with $\pi_{z_{d,t-1}, z_{d,t}}^p \geq u_{d,t}$ are considered when we sample $z_{d,t}$ that transits out of $z_{d,t-1}$.

Given the auxiliary variables $u_{d,t=1:T}$, the sequence $z_{d,t=1:T}$ can be updated through a forward-backward procedure. The forward message is

$$\begin{aligned} &p(z_{d,t}|y_{d,1:t}, u_{d,1:t}, g_{d,t} = p, \pi^p, \boldsymbol{\theta}) \\ &\propto p(z_{d,t}, y_{d,t}, u_{d,t}|y_{d,1:t-1}, u_{d,1:t-1}, g_{d,t} = p, \pi^p, \boldsymbol{\theta}) \\ &= p(y_{d,t}|z_{d,t}, \boldsymbol{\theta}) \sum_{z_{d,t-1}:\pi_{z_{d,t-1}, z_{d,t}}^p > u_{d,t}} p(z_{d,t-1}|y_{d,1:t-1}, u_{d,1:t-1}, g_{d,t} = p, \pi^p) \end{aligned} \tag{18}$$

where the first term can be estimated as

$$p(y_{d,t}|z_{d,t}, \boldsymbol{\theta}) = \sum_{j=1}^{J_k} \frac{n_{k,j}^{-t}}{\gamma^* + n_{k.}^{-t}}p(y_{d,t}|\theta_{k,j}) \quad + \frac{\gamma^*}{\gamma^* + n_{k.}^{-t}}p(y_{d,t}|\phi_k, S_0, d_0) \tag{19}$$

and the sequence $z_{d,t=1:T}$ can be sampled through a backward pass

$$p(z_{d,t}|z_{d,t+1}, y_{d,1:T}, u_{d,1:T}, g_{d,t}, \boldsymbol{\pi}, \boldsymbol{\theta}) \propto \quad p(z_{d,t}|y_{d,1:t}, u_{d,1:t}, \boldsymbol{\theta})p(z_{d,t+1}|z_{d,t}, u_{d,t+1}, g_{d,t} = p, \pi^p) \tag{20}$$

### B.1.2 Sampling $\pi^p$

To sample the eye movement state transition matrix $\pi^p$, let $n_{kk'}^p$ denote the total occurrences of transitions from pattern $k$ to $k'$ at phase $p$. The conditional distribution of transition matrix $\pi^p$ is

$$(\pi_{k1}^p...\pi_{kK}^p, \sum_{k'=K+1}^{\infty} \pi_{kk'}^p) \sim \quad Dir(n_{k1}^p + \alpha\beta_1, ...n_{kK}^p + \alpha\beta_K, \alpha\beta_u) \tag{21}$$

where $\beta_u = \sum_{k'=K+1}^{\infty} \beta_{k'}$ and $K$ is the total number of instantiated states. $\boldsymbol{\beta}$ is sampled from

$$(\beta_1, ...\beta_K, \beta_u) \sim Dir(m_{.1}, ..., m_{.K}, \gamma) \tag{22}$$

where $m_{.k} = \sum_p \sum_{k'} m_{k'k}^p$ and based on [41]

$$p(m_{k'k}^p = m|z, \alpha, \boldsymbol{\beta}) \propto S(n_{k'k}^p, m)(\alpha\beta_k)^m \tag{23}$$

$S(.,.)$ is the Stirling number of the first kind.

### B.1.3 Sampling $\theta_{k,j}$

Given the eye movement pattern assignments $(z_{d,t}, s_{d,t})$, we sample the parameter of eye movement patterns $\theta_{k,j} = (A_{k,j}, \Sigma_{k,j})$. With the prior $(A_{k,j}, \Sigma_{k,j}) \sim NIW(S_0, d_0, \kappa, \phi_k)$, it can be shown that

$$
\begin{aligned}
p(A_{k,j}, \Sigma_{k,j}|Y_{k,j}, \phi_k) &= NIW(S^*, d_0 + n_{k,j}, \kappa + n_{k,j}, A^*) \\
A^* &= \frac{\kappa}{\kappa + n_{k,j}}\phi_k + \frac{n_{k,j}}{\kappa + n_{k,j}}\bar{y} \\
S^* &= S_0 + \sum_{d,t:z_{d,t}=k,s_{d,t}=j}(y_{d,t}-\bar{y})(y_{d,t}-\bar{y})' + \frac{\kappa n_{k,j}}{\kappa + n_{k,j}}(\bar{y}-\phi_k)(\bar{y}-\phi_k)'
\end{aligned}
\tag{24}
$$

Similarly, $\phi_k$ is updated as $p(\phi_k|\theta_{k.}) = N(\phi^*, U^*)$ with

$$U^* = (U_0^{-1} + \sum_j \Sigma_{k,j}^{-1}/\kappa)^{-1} \quad \phi^* = U^*(\sum_j (\Sigma_{k,j}/\kappa)^{-1}\theta_{k,j} + U_0^{-1}A_0) \tag{25}$$

## B.2 Sampling Latent Variables of Verbal Narrations

### B.2.1 Sampling $v_{d,t}$

For sampling the topic assignments, let $n_{p,l}$ denote the occurrence of words corresponding to topic $l$ at phase $p$, and $\rho_{l,w}$ denote the weight of word $w$ in topic $l$. Given phase $g_{d,t}$, hidden topic assignment $v_{d,t}$ is sampled from

$$p(v_{d,t} = l|\boldsymbol{v}^{-dt}, g_{d,t} = p, \boldsymbol{\tau}, \boldsymbol{\rho}) \propto \begin{cases} (n_{p,l} + a\zeta_l)\rho_{l,w} & l \leq L \\ a\zeta_u\omega & l = L+1 \end{cases} \tag{26}$$

where $L$ is the number of instantiated topics and $n_{p,l}$ is number of words assigned to topic $l$ at phase $p$.

### B.2.2 Sampling $\tau_p$

The topic distribution $\tau_p$ is sampled from

$$\tau_p = (\tau_{p,1}, ..., \tau_{p,L}, \tau_{p,u}) \sim Dir(n_{p,1} + \alpha\zeta_1, ..., n_{p,L} + \alpha\zeta_L, \alpha\zeta_u) \tag{27}$$

where $\tau_{p,u} = \sum_{l'=L+1}^{\infty} \tau_{p,l'}$ and $L$ is the number of instantiated topics. $\zeta$ is sampled from

$$(\zeta_1, ..., \zeta_L, \zeta_u) \sim Dir(m_{.1}, ..., m_{.L}, \eta) \tag{28}$$

where $m_{.l} = \sum_p m_{p,l}$ and based on [41]

$$p(m_{p,l} = m|z, \alpha, \zeta) \propto S(n_{p,l}, m)(\alpha\zeta_l)^m \tag{29}$$

$S(.,.)$ is the Stirling number of the first kind.

### B.2.3 Sampling $\rho_l$

Given the topic assignments $\boldsymbol{v}$, the word weights in topic $l$ are sampled from

$$\rho_l \sim Dir(\omega + \sum_d \sum_t \delta(w_{d,t} = w, v_{d,t} = l)) \tag{30}$$

### B.3 Sampling Latent Variables of Decision Phases

#### B.3.1 Sampling $g_{d,t}$

The phase assignments $g_{d,t=1:T}$ for time step $t = 1 : T$ in sequence $d$ assumes a first-order Markov structure, and can be block sampled through a forward-backward procedure given the state assignments $\boldsymbol{z}_d$ and topic assignments $\boldsymbol{v}_d$. Note that

$$
\begin{aligned}
p(g_{d,1:T}|\psi, \boldsymbol{v}_d, \boldsymbol{z}_d, \boldsymbol{\tau}, \boldsymbol{\pi}) =& p(g_{d,T}|g_{d,T-1}, \psi, \boldsymbol{v}_d, \boldsymbol{z}_d, \boldsymbol{\tau}, \boldsymbol{\pi}) \\
& p(g_{d,T-1}|g_{d,T-2}, \psi, \boldsymbol{v}_d, \boldsymbol{z}_d, \boldsymbol{\tau}, \boldsymbol{\pi})...p(g_{d,1}|\psi, \boldsymbol{v}_d, \boldsymbol{z}_d, \boldsymbol{\tau}, \boldsymbol{\pi})
\end{aligned}
\tag{31}
$$

So the sampling process starts by calculating the backward message from time step $t = T$ to $t = 1$, then sampling the latent phases from $t = 1$ to $t = T$. The backward message is defined as

$$
\text{msg}_{t,t-1}(g_{d,t-1} = p) \propto \begin{cases} \sum_{g_{d,t}} p(g_{d,t}|\psi)p(z_{d,t}|\pi^p)p(v_{d,t}|\tau_p)\text{msg}_{t+1,t}(g_{d,t}) & t \leq T \\ 1 & t = T+1 \end{cases}
\tag{32}
$$

$$
\propto p(v_{d,t:T}, z_{d,t:T}|g_{d,t-1}, \psi, \boldsymbol{\tau}, \boldsymbol{\pi})
$$

The phase assignment $g_{d,t}$ is sampled in a forward pass as

$$
p(g_{d,t} = p|\psi, \boldsymbol{v}_d, \boldsymbol{z}_d, \boldsymbol{\tau}, \boldsymbol{\pi}) \propto p(g_{d,t}|\psi_{g_{d,t-1}})p(v_{d,t}|\tau_p)p(z_{d,t}|\pi^p)\text{msg}_{t+1,t}(g_{d,t})
\tag{33}
$$

#### B.3.2 Sampling $\psi$

Each row of the phase transition matrix $\psi$ is updated as

$$
\psi_p \sim Dir(n_{p1} + \omega_0, ..., n_{pp'} + \omega_0..., n_{pP} + \omega_0)
\tag{34}
$$

where $n_{pp'}$ is the counts of transition from phase $p$ to $p'$.

### B.4 Algorithm for Posterior Inference

Algorithm 1 describes the MCMC sampling for posterior inference. The split-merge-switch sampler, which is discussed in the next subsection, is applied in each iteration for fast mixing.

### B.5 The Split-Merge-Switch Sampler

To make the posterior inference more efficient, a Split-Merge-Switch (SMS) sampler is applied along with the sampling algorithm discussed above.

The SMS sampler considers three types of proposals, namely *split, merge, and switch*. The three proposals are mutually exclusive. The proposal is evaluated using a Metropolis-Hasting acceptance ratio. If accepted, the proposal is implemented; if not, we ignore the proposal and proceed to sample other variables.

#### B.5.1 Mutual Exclusivity

In this section, we show that the three proposals in the SMS sampler are mutually exclusive. This property demonstrates the difference between the proposed SMS sampler and the existing split-merge sampler which is designed for a flat cluster structure. The switch move, which is designed for *non-parametric hierarchical clustering*, allows the SMS sampler to converge faster, as evidenced by our experimental results reported in Appendix D.

The split, merge, and switch proposals in the SMS sampler are mutually exclusive, i.e., none proposal can be achieved by performing a series of other proposals.

Let $F_a(S)$ denote a series of split and merge moves starting from $S$, $F_b(S)$ denote a series of merge and switch moves starting from $S$, and $F_c(S_i, S_e)$ denote a series of split and switch moves starting from $S_i$ and $S_e$ Suppose to the contrary that

$$
\exists S : \text{Switch}(S) = F_a(S)
$$

---
**Algorithm 1** The Posterior Inference Process
---
randomly initialize parameters $\rho$, $\theta$, $\phi$ and assignments $g$, $v$, $z$, $s$;
**for** iteration $i = 1 : maxIter$ **do**
  **for** sequence $d = 1 : D$ **do**
    **for** time step $t = 1 : T$ **do**
      update phase assignment $g_{d,t}$ using (33);
    **end for**
    update parameter $\psi$ using (34);
  **end for**
  **for** sequence $d = 1 : D$ **do**
    **for** time step $t = 1 : T$ **do**
      update topic assignment $v_{d,t}$ using (26);
    **end for**
  **end for**
  **for** instantiated topic $l = 1 : L$ **do**
    update parameter $\rho_l$ using (30);
  **end for**
  **for** sequence $d = 1 : D$ **do**
    **for** time step $t = 1 : T$ **do**
      update state assignment $z_{d,t}$ using (20);
      update component assignment $s_{d,t}$ using (16);
    **end for**
  **end for**
  **for** instantiated state $k = 1 : K$ **do**
    **for** instantiated mixture component $j = 1 : J$ **do**
      update parameter $\theta_{k,j}$ using (24);
    **end for**
    update parameter $\phi_k$ using (25);
  **end for**
  update parameter $\boldsymbol{\pi}$ using (21);
  apply merge-split-switch sampler as discussed in Algorithm 2;
**end for**
---

By definition of switch proposal, $z_{d,t}^{new} \neq z_{d,t}^{old}, \quad \exists (d,t) \in \text{Switch}(S)$. However, by definition of the split and merge proposals $z_{d,t}^{new} = z_{d,t}^{old}, \quad \forall (d,t) \in F_a(S)$. Therefore $S$ does not exist.

Suppose to the contrary that

$$\exists S : \text{Split}(S) = F_b(S)$$

However, $\text{Split}(S)$ ends up with 2 mixture components, $F_b(S)$ ends up with 1 mixture component. Therefore $S$ does not exist.

Suppose to the contrary that

$$\exists S_i, S_e : \text{Merge}(S_i, S_e) = F_c(S_i, S_e)$$

However, $\text{Merge}(S_i, S_e)$ ends up with 1 mixture component, $F_c(S_i, S_e)$ ends up with no less than 2 mixture components. Therefore $S_i, S_e$ do not exist.

### B.5.2 The Split Proposal

The split proposal suggests splitting a mixture component into two within the same state. Let $S$ denote the indices of events assigned to state $k$ and component $j$. First, a pair of indices $[(d_i, t_i), (d_o, t_o)]$ is randomly selected from $S$ and serve as anchors. Then we remove them from $S$ and form singleton sets $S_i = \{(d_i, t_i)\}$ and $S_o = \{(d_o, t_o)\}$. For each $(d,t) \in S$, we sequentially add it to $S_i$ with

$$p((d,t) \in S_i | S_i, S_o, y_{d,t}) = \frac{|S_i| \int f(y_{d,t}|\theta) dH_{S_i}(\theta)}{|S_i| \int f(y_{d,t}|\theta) dH_{S_i}(\theta) + |S_o| \int f(y_{d,t}|\theta) dH_{S_o}(\theta)} \tag{35}$$

where $H_{S_i}(\theta)$ is the posterior distribution of $\theta$ based on the prior $p(\theta|\phi)$ and the current members in $S_i$, and $f(y_{d,t}|\theta)$ is the probability of $y_{d,t}$ conditioned on its pattern assignment and $\theta$.

Otherwise, we add it to $S_o$.

The general form of Metropolis-Hastings acceptance ratio is provided as

$$a(\boldsymbol{\eta}^{old}, \boldsymbol{\eta}^{new}) = \min[1, \frac{p(\boldsymbol{\eta}^{new})p(\boldsymbol{y}|\boldsymbol{\eta}^{new})p(\boldsymbol{\eta}^{old}|\boldsymbol{\eta}^{new})}{p(\boldsymbol{\eta}^{old})p(\boldsymbol{y}|\boldsymbol{\eta}^{old})p(\boldsymbol{\eta}^{new}|\boldsymbol{\eta}^{old})}] \tag{36}$$

where $\boldsymbol{\eta}^{old}$ denotes the old assignments, $\boldsymbol{\eta}^{new}$ denotes the proposed new assignments. It satisfied the detailed balance, which guarantees the convergence of the MCMC sampler [42].

For a split proposal, $p(\boldsymbol{\eta}^{old}|\boldsymbol{\eta}^{new})$ is always 1, because there is only one way to transit from $\boldsymbol{\eta}^{new}$ to $\boldsymbol{\eta}^{old}$. $p(\boldsymbol{\eta}^{new}|\boldsymbol{\eta}^{old})$ is the probability of assigning members from $\boldsymbol{\eta}^{old}$ to the split allocations $\boldsymbol{\eta}^{new}$, which equals to the product of $p((d,t)|S_i, S_o, y_{d,t})$ for all $(d,t)$ based on the new allocation.

$p(\boldsymbol{\eta}^{new})$ and $p(\boldsymbol{\eta}^{old})$ are calculated by using the Polya's urn metaphor respectively [29], and the ratio $\frac{p(\boldsymbol{\eta}^{old})}{p(\boldsymbol{\eta}^{old})}$ is

$$\frac{p(\boldsymbol{\eta}^{new})}{p(\boldsymbol{\eta}^{old})} = \gamma^* \frac{(|S_i| - 1)!(|S_o| - 1)!}{(|S| - 1)!} \tag{37}$$

where $\gamma^*$ is the concentration parameter defined in (5). And the ratio $\frac{p(\boldsymbol{y}|\boldsymbol{\eta}^{new})}{p(\boldsymbol{y}|\boldsymbol{\eta}^{old})}$ is

$$\frac{p(\boldsymbol{y}|\boldsymbol{\eta}^{new})}{p(\boldsymbol{y}|\boldsymbol{\eta}^{old})} = \prod_{(d,t)\in S_i} \int f(y_{d,t}|\theta)dH_{S_i}(\theta) \prod_{(d,t)\in S_o} \int f(y_{d,t}|\theta)dH_{S_o}(\theta)$$
$$/ \prod_{(d,t)\in S} \int f(y_{d,t}|\theta)dH_S(\theta) \tag{38}$$

### B.5.3 The Merge Proposal

The merge proposal suggests merging two mixture components from the same state into one single one. It is essentially the reverse of a split proposal, and the acceptance ratio is calculated using (36).

$p(\boldsymbol{\eta}^{new}|\boldsymbol{\eta}^{old})$ is always 1 since there is only one way of merging, while $p(\boldsymbol{\eta}^{old}|\boldsymbol{\eta}^{new})$ is the joint probability of assigning members from the merged allocations $\boldsymbol{\eta}^{new}$ to the old allocations $\boldsymbol{\eta}^{old}$, which equals to the product of $p((d,t)|S_i, S_o, y_{d,t})$ for all $(d,t) \in S$ based on the old allocation.

For a merge proposal, the ratio the prior distribution is calculated in a reversed way to that in a split proposal.

$$\frac{p(\boldsymbol{\eta}^{new})}{p(\boldsymbol{\eta}^{old})} = \frac{1}{\gamma^*} \frac{(|S| - 1)!}{(|S_i| - 1)!(|S_o| - 1)!} \tag{39}$$

and the ratio of the likelihoods is calculated as

$$\frac{p(\boldsymbol{y}|\boldsymbol{\eta}^{new})}{p(\boldsymbol{y}|\boldsymbol{\eta}^{old})} = \prod_{(d,t)\in S} f(y_{d,t}, \theta)dH_S(\theta)$$
$$/ \left( \prod_{(d,t)\in S_i} \int f(y_{d,t}, \theta)dH_{S_i}(\theta) \prod_{(d,t)\in S_o} \int f(y_{d,t}, \theta)dH_{S_o}(\theta) \right) \tag{40}$$

### B.5.4 The Switch Proposal

The switch proposal suggests moving some mixture components from one state and adding them to another state as additional mixture components, while keeping the grouping of elements within each mixture unchanged. We further consider two cases, namely switch1, where some mixture components from a pattern $k_1$ are moved to a new pattern $k_2$, and switch2 where all mixture components from a pattern $k_3$ are moved to an existing pattern $k_4$. For switch1, we randomly sample a pair of mixture component indices from pattern $k_1$ and serve as anchors. Then we sequentially allocate other mixture components in a similar way to Eq 35, and calculate the probability of the proposal. The acceptance rate is calculated as Eq 36. However, the switch incurs the change of both pattern assignment and mixture assignment,

$$\frac{p(\boldsymbol{\eta}^{new})}{p(\boldsymbol{\eta}^{old})} = \frac{p(\boldsymbol{s}^{new}|\boldsymbol{z}^{new})}{p(\boldsymbol{s}^{old}|\boldsymbol{z}^{old})} \frac{p(\boldsymbol{z}^{new})}{p(\boldsymbol{z}^{old})} \tag{41}$$

For all mixture components nested in a pattern $k$,

$$p(\boldsymbol{s}|z = k) = (\gamma^*)^J \frac{\prod_j (n_{k,j} - 1)!}{\prod_{i=1}^{n_k}(\gamma^* + i + 1)} \tag{42}$$

Therefore,

$$\frac{p(\boldsymbol{s}^{new}|\boldsymbol{z}^{new})}{p(\boldsymbol{s}^{old}|\boldsymbol{z}^{old})} = \prod_{i=1}^{|S_2|}(\gamma^* + i - 1) \prod_{i=1}^{|S_3|}(\gamma^* + i - 1)/\prod_{i=1}^{|S_1|}(\gamma^* + i - 1) \tag{43}$$

And according to [32],

$$p(\boldsymbol{z}^{old}) = \gamma^K \beta_{u_1}\beta_{u_2}...\beta_{u_k} \prod_{k=1}^{K}\prod_{k'=1}^{K}\prod_{i=1}^{n_{kk'}}(\gamma\beta_{k'} + i - 1)/\prod_{k=1}^{K}\prod_{i=1}^{n_k}(\gamma + i - 1) \tag{44}$$

where $\beta_{u_k} = 1 - \sum_{i}^{k-1}\beta_i$. And $p(\boldsymbol{z}^{new})$ can be calculated similarly.

The switch2 proposal is essentially the reverse of a switch1 proposal, and the acceptance ratio is calculated in the reversed way. Notice that the switch also incurs the change of corresponding $\boldsymbol{\theta}$'s conditional dependency on $\boldsymbol{\phi}$, because the component is assigned to a new state.

Algorithm 2 describes the split-merge-switch sampling.

---

**Algorithm 2** The Split-Merge-Switch Sampler

---

randomly draw a data point indexed by $i$, and record its pattern assignment $z$. From the set of data points assigned $z$, draw another data point indexed by $o$. Use $(i, o)$ as anchors;
**if** $s_i = s_o$ **then**
    propose a split and calculate acceptance ratio $AR$ using (36), (37), (38);
    accept the proposal with $AR$ and update mixture assignments;
    resample stick portion $b_{z_i}$ and $b_{z_o}$ after the split;
**else**
    propose a merge and calculate acceptance ratio $AR$ using (36), (39), (40);
    accept the proposal with $AR$ and update mixture assignments;
    resample stick portion $b_{z_i}$ after the merge;
**end if**
randomly draw a data point indexed by $i$, and record its pattern and mixture assignment $(z, s)$. From the set of data points not assigned as $(z, s)$, draw another data point indexed by $o$. Use $(i, o)$ as anchors;
**if** $z_i = z_o$ **then**
    propose a switch1 proposal;
**else**
    propose a switch2 proposal;
**end if**
calculate acceptance ratio $AR$ using (36) (41);
accept the proposal with $AR$ and update state assignments;
resample stick portion $\beta$ and and transition matrix $\pi$ after the switch;

---

# C Additional Details of Experiments

In this section, we provide details of experiment settings and additional results.

## C.1 Experiment Settings

The hyper-parameters are: $\gamma = 1/20$, $\gamma^* = 1/100$, $\alpha = 1/16$, $\xi = 1/8$, $a = 1/16$, $\omega = 1/40$, $\kappa = 0.8$, $A_0 = 0$, $U_0 = 100 * I$, where $I$ denotes the identity matrix.

For fusing eye movements data and verbal narrations, the unsupervised model is trained using the entire data sets. For downstream supervised tasks including the disease morphology classification and diagnostic correctness prediction, eighty percent of the data is randomly split for training, and the rest is used for testing. Five-fold cross-validation is applied for hyper-parameter tuning. For model training, we only use the features of eye fixations. We use saccade features mainly for interpretation: 1) We draw lines for saccades to visualize eye movement patterns; 2) We interpret patterns as concentration/switching/cluttering using fixation duration and saccade amplitude.

Table 4: Inferred Topics with Most Frequent Words (The topic indexes in this table match the topic indexes in Fig 4 from the main paper)

| Topic | Top Words | | | | | | | |
|---|---|---|---|---|---|---|---|---|
| 1 | there | looks | appears | so | little | see | some | erythematous |
| 2 | upper | chest | back | patches | hypopigmented | neck | areas | hyperpigmented |
| 3 | nose | left | nasal | telangiectasias | child | tip | young | overlying |
| 4 | erythema | foot | there's | lower | some | crusting | there | dorsal |
| 5 | erythematous | plaques | scale | patches | some | annular | plaque | scaly |
| 6 | papules | lesions | these | some | multiple | on | elbow | scattered |
| 7 | papule | nodule | there's | lesion | brown | center | surrounding | on |
| 8 | macules | as | papules | some | they | scattered | three | they're |
| 9 | vesicles | in | bulla | tense | there's | bullae | well | as |
| 10 | patch | right | lip | lower | area | depigmented | serpiginous | extending |
| 11 | on | like | or | it's | what | skin | bit | not |
| 12 | could | differential | versus | think | my | diagnosis | would | also |
| 13 | would | or | other | would | like | as | probably | in |
| 14 | vitiligo | vasculitis | scleroderma | would | xanthomas | leukocytoclastic | xanthoma | petechiae |
| 15 | lupus | drug | eruption | would | multiforme | cutaneous | reaction | erythematosus |
| 16 | tinea | corporis | mycosis | erythema | fungoides | annulare | versicolor | would |
| 17 | melanoma | vascular | malignant | blue | sort | kaposi's | lesion | nodular |
| 18 | lichen | linear | planus | striatus | epidermal | nevus | see | scabes |
| 19 | seborrheic | nevus | post-inflammatory | keratosis | hypopigmentation | halo | morphea | diagnosis |
| 20 | bullous | pemphigoid | pemphigus | linear | vulgaris | drug | disease | blistering |
| 21 | or | could | folliculitis | infection | like | acne | varicella | zoster |
| 22 | but | it's | so | have | not | think | if | don't |
| 23 | larva | cutaneous | migrans | cafe-au-lait | neurofibromatosis | like | hyperpigmentation | one |
| 24 | dermatitis | contact | allergic | eczematous | eczema | acute | can | rosacea |
| 25 | cell | basal | carcinoma | nevus | pigmented | tumor | spitz | juvenile |
| 26 | but | would | my | for | one | number | at | more |
| 27 | say | i'm | about | gonna | would | am | only | for |
| 28 | percent | certainty | final | diagnosis | sure | fifty | go | ninety |

## C.2 Additional Results of Interred Phases and Cross-Modal Interactions

Additional Results of phase transition, state transition, and topic distribution of each phase are provided in Figures 6 and 7.

Each phase has a strong tendency of self transition, and there are moderate occurrences of phase transition from description (P1) to reasoning (P2), and from reasoning to conclusion (P3). Besides, each phase is associated with a unique set of topics (e.g., the description phase is closely related to topics T3, T4, and the conclusion phase is closely related to topics T28, T31).

Figure 6: Visualization of Decision Phase Transition Matrix (Left) and Topic Distribution of Each Phase (Right) in Experiment II

For eye movements, we observe higher self-transition of S10(concentration) at the reasoning and conclusion phase, which indicates that physicians' eye movements are more stable upon conclusion. There are more non-concentration transitions at the description phase, such as the transition from S13 to S5(clutter), indicating that experts change eye movement states more frequently at the description phase in order to gather general information from the entire image.

Figure 7: Visualization of Eye Movement State Transition Matrix at Description Phase (Left), Reasoning Phase (Middle) and Conclusion Phase (Right) in Experiment II

## C.3 Top Words from Inferred Topics

The inferred topics with top words from each topic are summarized in Table 4. The inferred topics provide rich information about the diagnostic decision-making process of physicians. For instance, Topic 7 mainly correspond to disease morphology, ('papule' and 'nodule','lesion'); Topic 2 correspond to body location ('chest', 'back', 'neck'). Therefore, they are closely related to the description phase. Topics T12 is closely related to the reasoning phase, as they contain keywords that describe a physician's logical thinking (i.e., differential, versus, think). Topics T28 is closely related to the conclusion phase, as they contain keywords about a diagnostic decision (i.e., certainty, final).

## C.4 Visualization of Eye Movement Patterns

Additional inferred eye movement patterns, including states (main patterns) and mixture components (sub-patterns), are summarized in Table 5. Notice that we use terms including concentration, switching and clutter only for interpretating the eye movement patterns, while the main and sub-patterns discovered by the nonparametric model are based on statistical regularities. Eye movement patterns usually match the disease morphology, and thus the patterns are usually similar for the same disease. For instance, the skin disease *Halo Nevus* is usually characterized by a single discrete lesion (morphology), and the concentration pattern dominates eye movement patterns for most experts who viewed the image. *Phytophotodermatitis* is usually characterized by diffuse lesions with random distribution, and the clutter pattern dominates eye movement patterns.

Table 5: Visualization of Selective Eye Movement Patterns: Each row corresponds to a state (main pattern), each column corresponds to a mixture component (fine-grained pattern) and each image is an example of the pattern

The inferred states (main patterns) capture high-level characteristics of eye movements. For instance, State 1 can be interpreted as clutter pattern because it is characterized by saccades with large amplitude; State 2 can be interpreted as concentration because it is characterized by fixations within a small area, and State 3 can be interpreted as switching because it is characterized by saccade between

two locations. Different states correspond to different characteristics. For instance, both States 2 and 7 can be interpreted as concentration, but State 7 manifests slightly larger fixed areas than State 2.

A state may have multiple components (sub-patterns), and those components are slightly different from each other. For instance, State 2 has three components. All those components can be interpreted as concentration, but Component 3 manifests slightly larger fixed areas than Component 1.

# D  Ablation Study on Split-Merge-Switch Sampler

For illustration purpose, we perform an ablation study on the SMS sampler using synthetic data where the structure of the latent states are known in advance, which can serve as ground-truth for evaluation.

The synthetic data in the ablation study provides a controlled experiment environment where the ground truth of the number of main patterns and sub-patterns are known. It is generated through hierarchical Gaussian mixtures with four main patterns and 16 sub-patterns in total. The main patterns are centered at [4.5,4.5], [-4.5,4.5], [-4.5,-4.5], and [4.5,-4.5] respectively. Each main pattern has 4 sub-patterns, whose center deviates from its main pattern's center by [1.5 1.5], [-1.5,1.5], [-1.5,-1.5], [1.5,-1.5], as shown in Figure 8. We generate 50 sequences, where each sequence has 50 observations. The first observation of a sequence is randomly assigned a main pattern and a sub-pattern, while the following observations' pattern assignments are drawn from a pre-set Markov transition matrix. Given the pattern assignments, each observation is drawn from a Gaussian distribution $N(\theta_{k,j}, 0.5 \times I)$, where $I$ is an identity matrix.

Figure 8: Visualization of sub-patterns: ground truth (left) and random initialization (right)

We initialize the number of main patterns $K = 2$ and the number of sub-patterns in each main pattern $J = 2$. The pattern assignments for all observations are randomly initialized, as shown in Figure 8. An infinite hidden Markov Model with nested Dirichlet process (iHMM-nDP) algorithm is implemented, with or without the SMS sampler.

Figure 9: Visualization of inferred main patterns with and without SMS sampler (Left and Middle), Log marginal likelihood up to 500 iterations (Right)

The inferred main patterns are visualized in Figure 9. To quantitatively analyze how the SMS sampler affects the performance, we estimate the log marginal likelihood for one run with or without the SMS sampler. If SMS sampler is not applied, the Beam sampler, the classical split-merge (SM) sampler [29], and the block sampler (with truncation level set to 4) are used as backbones. Results indicate that the SMS sampler contributes to the fast mixing rate and better hierarchical clustering results.