[Reviews · NeurIPS 2020]

Review 1

Summary and Contributions: The paper proposed a dynamic data fusion for complex decision-making. The authors proposed a novel model that allows the latent knowledge patterns of perceptual behavior and conceptual processing to be assigned during the key phases of the decision-making process.

Strengths: The topics covered in this paper are novel and the proposed method is also new. The paper is also well written overall.

Weaknesses: The paper mention that extensive post-processing needed in order to get the results. How much time does it take to produce the results? Could the eye-tracking data be just as useful with post-processing as the combined method? The phase-specific words are questionable and not very well defined. The experimental conditions are not very clear and for example in Table 1, Part 1, Image 1 the word "like" is not highlighted. Eye-tracking data is often noisy and the description of each image can vary heavily based on the expert as well. It is not clear to me if the authors also performed experiments on different diseases as well as multiple representations of the same disease and if they compared the characteristics of the eye movements between those images.

Correctness: I am sceptical about the verbal feedbacks of the experts and their usefulness as well as the selected keywords which meant to determine the phase of the decision-making procedure as well as the reliability of gaze patterns when it comes to varying size and appearance of symptoms of different diseases.

Clarity: The paper is well written, along with the supplementary material one can understand the notations, lemmas, and algorithms. However, some indexes are not correct to see Supplementary C2 mentions topic 29,30,32, however, they are not included in Table 4.

Relation to Prior Work: Yes

Reproducibility: Yes

Additional Feedback: I would like to understand how controlled the experimental environment was, how useful this algorithm would be in a real-world scenario where the experts are not following a scheme while providing feedback rather just describing the given image freely. How the gaze patterns are similar when we show different cases of the same disease.


Review 2

Summary and Contributions: The paper proposes a new model for nonparametric hierarchical clustering of multimodal (eye movement and narration) data. It additionally gives a new sampler for this complex model that has better mixing than other samplers applied to this domain. Finally, it evaluates the model on naturalistic data from expert dermatologists, using the model to discover structure in the data and generate predictions both of the quality of the experts' decisions, and of properties of the actual stimulus the experts are evaluating.

Strengths: The model is sophisticated with many moving pieces, but these pieces are both well-motivated and justified in improved performance relative to competitor models. In particular, the structure of the model (the simultaneous nonparametric clustering by topic as well as temporal segmentation to phases) exploits insights about the specific domain in question, and creates an interpretable structure which is often necessary in convincing experts to leverage ML-based decision support. For evaluation, benefits of the sampler are shown separately from benefits of the model, comparisons are given from both multimodal and unimodal aspects, and empirical evaluation on real data tackles two distinct problems (validation of expert judgments, and prediction in the absence of expert judgement).

Weaknesses: From an eye movement and cognitive processing perspective, domain knowledge could be exploited better. In particular, saccades are stereotyped and ballistic, so given a pair of fixations I can make a pretty good prediction of the saccade amplitude and velocity. In light of this the model should (at least theoretically) perform similarly well with less data by ignoring saccades and using the fixations alone, and I am a bit puzzled by the use of saccade features in addition to fixations. In addition, the paper would be stronger in empirically demonstrating that the best number of reasoning phases is indeed 3, by e.g. comparing performance against models with more / fewer reasoning phases. If so, we may discover that the best segmentation of the experts' process is different than that proposed. Beyond this, evaluation on other visual narration domains beyond just dermatology would strengthen the paper. So would addressing some of the issues brought up below re: correctness, clarity, and relation to prior work.

Correctness: The paper doesn't make it obvious why lemmas 1 and 2 should be true (i.e. why the acceptance ratios satisfy detailed balance and therefore the sampler is valid). No proof is given in the main text or the appendix as far as I can tell. I have the rough intuition that mutual exclusivity of the proposals means that the merge and split proposals satisfy detailed balance by being reverses of each other, and the switch proposal satisfies it by itself, but this could be made more explicit.

Clarity: Largely yes. It could still use another editing pass, for example: line 160: different trails -> different trials? not sure what is intended here line 165: AR-iMM -> AR-iHMM line 203 time-confusing -> time consuming line 295 "In most cases, the proposed model achieves best performance" -- according to table 2 this is all cases tested, right? So can just say "in all cases tested, the proposed model achieves best performance". Also, there are many distinct new terms introduced (phase, state, mixture component, semantic vs perceptual components, etc). For the most part they are explained and the paper reminds the reader of their meaning when they are reused, but the extensive usage of italics both for defining terms and for emphasis makes it confusing whether something is intended as one or the other (and in general italics for emphasis are overused relative to what one typically sees in papers). The paper may be clearer if italics were reserved for definition of terms, and/or if emphasis were switched from italics to bold text or similar.

Relation to Prior Work: The baseline/competitor models are fairly well-chosen, but could be better-connected to the current contribution. For example, in the discussion of the eye movement models the paper deliniates contributions relative to the classical HDP-HMM model, but the unimodal eye movement baseline is a MAR model. So the paper would benefit from either explicit comparison to a HDP-HMM baseline, or simply reminding the reader the relationship between the eye movement model in the novel contribution and the one in the baseline (which I think exists -- wouldn't removing the hierarchy and nonparametric prior from the provided model yield MAR?). I think the unimodal narration baseline is closer, in the sense that the narration component of the model in the paper seems to be LDA with extra phase level-hierarchy and the comparison model is LDA -- but this connection is not made.

Reproducibility: Yes

Additional Feedback: UPDATE: I have read the rebuttal and the other reviewers' comments. I appreciate the rebuttal's clarification about the sampler's properties and the empirical evaluation of number of reasoning phases used. The paper is still missing some broader impact on the community to be in the top of accepted papers (e.g. going beyond dermatologists for the data, beyond applying the sampler to a single model, etc), though I recognize that there is so much content there that there's probably no room for more. I still think that this is a solid paper and I hope to see it at NeurIPS. Nitpicking: why say MN prior in line 190 and that area even though column cov is I? This is just a MVN prior separably on each column.


Review 3

Summary and Contributions: POST-REBUTTAL: I would like to thank the authors for their rebuttal and detailed responses to the comments. The interpretation of the words and topics is very helpful. I remain positive about this paper and would argue for acceptance. ================ This paper proposes an approach to model verbal narration and eye movements of clinicians in order to discover patterns to help understand the decision-making patterns. The authors propose a dynamic Bayesian non-parametric model and analyze two datasets of clinicians. The propose a split-merge-switch sampler to help efficiently explore the posterior state space.

Strengths: The paper tackles an interesting multi-modal task of combining verbal narrations with eye-tracking data during the analysis of images. The applications in the context of human AI collaboration and decision-understanding are clear. I am not aware of similar work in this space; however, I admit I am not a expert in the particular domain. But based on my research the novelty in this work appears sufficient. The paper is well written and clear. It is clear that there is a very large amount of content in this paper.

Weaknesses: I would have liked to have seen more examples in the discussion of the topics that were detected. It would be helpful if, in Table 1 and other similar illustrations the different topics that the colored words correspond to where explicitly indicated. In the supplementary material the table showing topics (Table 4) is useful, but I am curious to understand more about the links between the works in each topic category. Regarding baselines, I realize in multimodal problems, especially those using modalities that are frequently not employed (e.g., eye tracking) it is difficult to find state of the art models that are appropriate. So this is not a major criticism but it does feel that perhaps the justification of the chosen baselines could be added to. In Figure 3 the differences seem quite small and it is difficult to gauge the significance of the differences in the off-diagonals. I don't have a concrete recommendation here, but given the authors have so much content to fit into this paper, I might suggest they could use this space in another way. As mentioned above, there is a very large amount of content in this paper, which is good, but a possible criticism is that some sections, in particular the discussion feels a little limited. The broader impacts statement hasn't really made an effort to consider how a bad actor or careless practitioner could mis-/or irresponsibly use these methods. I doesn't take much creativity to think of such a scenario and I think it is a missed opportunity to attempt to address that.

Correctness: The methodology appears correct and the notation seems clear and consistent. I will note that I was not able to completely review the mathematics in the supplementary material in detail.

Clarity: The paper is well written.

Relation to Prior Work: I feel the related work is sufficient. However, with interdisciplinary work such as this, I will admit I am not familiar with all the subject areas - in particular topic modeling.

Reproducibility: Yes

Additional Feedback: Yes, I believe the method is reproducible but the dataset is would not be available to reproduce the results.


Review 4

Summary and Contributions: The paper proposes a novel dynamic Bayesian nonparametric model, called AR-iHMM-nDP, to jointly analyze the eye movements and verbal narrations to discover latent knowledge patterns of the decision making process. A new split-merge-switch sampler is proposed for posterior inference. Empirical results on diagnostic error prediction and disease morphology categorization were included to demonstrate the effectiveness of the proposed method.

Strengths: - The paper is well written, which helps people who are not familiar with the domain (i.e., multimodal knowledge data fusion) like me understand better. I particularly like the example in Figure 2, which is presented early in the paper and provides a nice illustration of the input data as well as the expected output of the model. - The model proposed AR-iHMM-nDP is novel. In particular, I find the main idea of the paper interesting: learning the key decision phases jointly from both eye movements and verbal narrations, in which each decision phase is characterized by (1) a distribution of word topics from the verbal narrations and (2) their dynamic interactions with eye movement patterns. - Using HDP prior to capture hierarchy of states, where the number of states are unknown, is statistically sound. - The paper also provides quite extensive analysis of the proposed Split-Merge-Switch (SMS) sampling algorithm for posterior inference, both on simulated and real datasets (most of which are included in the Appendix). - Extensive empirical results are included, both on real datasets (predictions of diagnostic correctness and prediction of disease morphology) and simulated data.

Weaknesses: The model is quite complicated and it'd be great if discussion on the scalability of the approach is included in more details. In addition, I'd also recommend including discussions on the size of the data, which seems small from the description in the paper and I'm not sure if such small datasets are sufficient to fit such complicated model. In addition, as discussed in the "Clarity" section, although I think the paper is overall well written, there are various notations and concepts which could be defined more clearly to make the graphical model and model description easier to follow.

Correctness: The proposed model and posterior inference sampler are sound to the best of my knowledge.

Clarity: Overall, I think the paper is expertly written. As mentioned above, I particular like the example (Fig. 2) which was placed early in the paper narrative and clearly illustrates the data. Here are some recommendations which might help make the paper clearer: - In Section 4.2, it was not clear to me what "main patterns", "sub-patterns", and "observations of eye movements" initially mean. And since all these concepts are not trivial, all the discussions on the three-level hierarchy as well as the latent states are quite hard to follow. - All the notations used in the graphical model and discussed throughout the paper are defined in Table 3, which is placed in the Appendix. I would recommend moving some of the key notations to the main paper. Otherwise, it's very difficult to understand the Fig. 1.

Relation to Prior Work: Reasonable baselines are considered and compared against in the experiments, both for the prediction tasks and the study on different posterior inference techniques.

Reproducibility: Yes

Additional Feedback:


Review 5

Summary and Contributions: The authors propose a model and framework to model decision-making process by using input eye movements and spoken (textual) narration. The paper presents a case-study of medical doctors observing images of skin abnormalities in order to diagnose them properly. The contributions of the paper include rather complex probabilistic model that is able to fuse both the gaze features and the textual narration and model the phases of the expert decision-making process. The authors use Bayesian approach and sampling and in order to have efficient solution they develop some new MCMC approaches to speed up sampling and convergence.

Strengths: The model is interesting and no doubt useful read for anyone wishing to combine gaze trajectories with text in order to model decision-making process. The authors have also tuned their sampler in order to be more efficient for topic models, which can be quite painful to sample from efficiently. The paper is quite concise and well written. The claims are sound and motivated and novel, as far as I can tell. The results are relevant for modelling human interaction in decision-making processes. Overall, I consider the paper quite good.

Weaknesses: The main weakness of the work is that it is a bit too concise, which makes parts of the manuscript challenging to follow. The authors have made a good job of compressing their contribution to the required page limits, but the contributions could easily have used more pages. Now some of the results etc. have been put to the supplementary material. The results are apparently mostly reproducible, but not fully (e.g., the data does not contain the words, only the indices, nor the competing algorithms were apparently not implemented). It would be better if the code could produce the results presented in the figures and tables of the paper. (I is not 100% clear for me which parts of the results are reproducible by the linked code; I did not actually test the code which needs software that I do not have, i.e., Matlab, to run.) The authors provide the code via a link in the supplementary material, but they do not state if and how they will publish it after possible acceptance.

Correctness: The paper is correctly written, as far as I can tell, and the empirical methodology appears to be valid. Due to the complexity of topic and conciseness of presentation I was unable to verify everything.

Clarity: The paper is written well, but as written above, the authors have struggled with the page limit.

Relation to Prior Work: Related work was properly discussed, to the extent to be expected from a conference paper.

Reproducibility: Yes

Additional Feedback: Overall, the paper was quite good and it managed to present a system for a relevant problem of modeling expert user's cognition by using gaze data and annotations. The only major issue with the paper I had was that the presentation was quite compressed due to page limits etc., and that the attached code could have been documented better and/or it could have been such that it had reproduced the tables and figures reported in the table. Minor typographic issues: "IRB approval": is this ethical review board (ERB?), please clarify. Please also consider adding space to units "50Hz" -> "50 Hz", "60cm" -> "60 cm". In references please write "monte carlo", "gibbs", "dirichlet" etc. with capital.

[Author Response · NeurIPS 2020]

We thank all the reviewers for the valuable comments/suggestions. We summarize our responses as follows:

**[Reviewer 2]** *Q1: Extensive post-processing.* We mentioned on page 2 that extensive post-processing is needed in reference paper [10], while our model addressed this issue by using a 3-level hierarchy to group semantically similar patterns automatically. We compared with MAR on eye-movement data (page 8), and our model outperforms MAR.

*Q2: Phase-specific words are not well defined.* The model assigns a topic to each word. In Table 1, we use different colors to highlight informative words from three topics (please refer to Section 5.2). A word is highlighted if its weight is high in the assigned topic. The second 'like' in Table 1 part 1 is not highlighted because the criterion is not satisfied.

*Q3: Eye movement patterns on different diseases and multiple representations of a disease.* Both Experiments I and II contain images of different types of dermatological diseases. Experiment II also contains multiple images for the same type of disease. Eye movement patterns usually match the disease morphology, and thus the patterns are usually similar for the same disease. For instance, the skin disease *Halo Nevus* is usually characterized by a single discrete lesion (morphology), and the concentration pattern dominates eye movement patterns for all experts who viewed the image. We will add illustrative examples to the revised paper.

*Q4: Experts not following a scheme.* In this work, we study knowledge data from experts who are trained professionals. They analyze the images in systematic process, and their verbal descriptions usually follow certain schemes. A future direction is to make the model more robust to cases where careless practitioners do not follow such schemes.

*Q5: Topics 29,30,32 are not included in Table 4.* Table 4 matches Figure 4 (please refer to Table 4's caption), which shows 28 topics inferred from Experiment I. The 32 topics are from Experiment II.

*Q6: How the controlled experimental environment was.* Please refer to Section 3 for the experimental environment.

**[Reviewer 3]** *Q1: The use of saccades.* We use saccade features mainly for interpretation: 1) We draw lines for saccades to visualize eye movement patterns; 2) We interpret patterns as concentration/switching/cluttering using fixation duration and saccade amplitude. For model training, we only use the features of fixations.

*Q2: Number of reasoning phases.* The classification accuracy of disease morphology for two experiments are 81.2 and 79.8 (2 phases), 78.5 and 77.1 (4 phases). For diagnostic correctness, the results are 70.9 and 68.5 (2 phases), 69.0 and 68.2 (4 phases). According to the results, the optimal number of phases is 3 (2-phase is the second-best).

*Q3: Why Lemma 1 and 2 are true.* Detailed balance guarantees the convergence of the MCMC sampler (please refer to reference paper [29]), and a general form of acceptance rate is given by Eq 12. Eq 14 is extended from Eq 12 for the switch proposals. Eqs 13 and 15 show how each component in the acceptance rates are calculated and are discussed in Appendix B.5. We will add more details about the proof in the revised paper.

*Q4: Mutual exclusivity of the proposals.* We would like to clarify that the mutual exclusivity is not used to prove the lemmas. Mutual exclusivity ensures that all three proposals are important for speeding up MC sampling, because the effect of one proposal cannot be substituted by the effect of a combination of other proposals.

*Q5: Relation to prior work.* We agree with the reviewer that for eye movements, removing the hierarchy and nonparametric prior yields MAR, and for narrations, removing the hierarchy yields LDA. We will add discussions.

*Q6: MN prior.* We agree that the prior is essentially multivariate normal (MVN) on each column. Since the posterior follows a matrix normal distribution (MN), we write the prior in the form of MN to be consistent.

**[Reviewer 4]** *Q1: More examples of the topics.* We will add interpretations of the words and topics in Table 4. For instance, words in Topic 1 mainly correspond to disease morphology, including color ('brown' and 'dark'), location ('area' and 'center'); words in Topic 2 correspond to patients and body location ('patient', 'nodule', 'foot', 'dorsal').

*Q2: Justification of baselines.* The baselines are closely related to our model in terms of pattern discovery (auto-regression for eye movements and topic models for narrations). Comparison with those baselines illustrates the benefits of our model: using a three-level hierarchy to leverage cross-modal interactions and improve latent pattern discovery.

*Q3: A careless practitioner could misuse the model.* Please refer to Reviewer 2 Q4.

**[Reviewer 5]** *Q1: Discussion of scalability.* We would like to clarify that our model is designed for knowledge data, which is usually on a small scale due to the cost of data collection from experts. Meanwhile, it is critical to incorporate domain knowledge for pattern discovery given the complex nature of these data. We collected 807 and 720 valid eye movement/narration sequences for the two experiments, which are in a medium scale for similar domains. The design of using phases, topics, and eye movement patterns effectively incorporate prior knowledge about the nature of data (i.e., there are 3 phases; each phase has its unique topic distribution) to discover useful/interpretable knowledge patterns.

*Q2: Terms in Section 4.2.* 'observations of eye movements' are the recorded eye-gaze locations; 'main patterns' and 'subpatterns' are inferred latent patterns that capture high- and low-level characteristics of eye movements.

**[Reviewer 7]** *Q1: Presentation & Code.* We will improve the presentation to highlight the main contribution. We will add the Java code for recovering the words from indices and visualization and improve the documentation. We plan to make the code public via Github if the paper is accepted.

*Q2: Terminologies/typos.* IRB is Institutional Review Board (similar to ERB). We will correct other typos as suggested.

[Meta-Review · NeurIPS 2020]

This paper has a lot of content: Interesting cognitive science question of modelling human decision-making, data fusion of texts and eye movements, modelled with a new dynamic Bayesian nonparametric model, and introduces a new sampler for the model. This paper received a special amount of attention, 5 reviews which were needed because the paper makes several different kinds of contributions. Hence it is not a stereotypical good conference paper having one neat idea and presenting convincing theoretical or empirical support for it. Reviewers discussed the paper intensively, concluding that the paper is likely to be interesting at NeurIPS, and since there is not easy fix to make it more suitable to the format such as dividing it into two papers, it is good enough to be accepted though not among the best papers. Clarity can easily be improved by the authors, and additional details added in both the paper and the supplement. The authors are requested to make the changes they point at themselves in their response.